# Odontogenic Sarcomas of the Mandible

**DOI:** 10.3390/biomedicines12030615

**Published:** 2024-03-08

**Authors:** Jared Akers, Emily Geisler, Suimin Qiu, Petros Konofaos, Hisham Marwan

**Affiliations:** 1Oral and Maxillofacial Surgery, The University of Texas Medical Branch, Galveston, TX 77555, USA; jaakers@utmb.edu; 2Plastics and Reconstructive Surgery, The University of Texas Medical Branch, Galveston, TX 77555, USA; elgeisle@utmb.edu (E.G.); pekonofa@utmb.edu (P.K.); 3Department of Pathology, The University of Texas Medical Branch, Galveston, TX 77555, USA; sqiu@utmb.edu

**Keywords:** sarcoma, head and neck, odontogenic, Maxillofacial

## Abstract

Odontogenic sarcomas are exceedingly rare and account for less than 5% of all Maxillofacial Sarcomas. It usually affects the younger population. The posterior mandible is the most commonly affected site. Radiographically, it appears as a large destructive radiolucent lesion with ill-defined margins. Histopathological diagnosis is usually difficult. Surgery is the mainstay treatment. The role of chemotherapy and radiation therapy is not clear. Here, we present a case study of a 30-year-old female patient diagnosed with odontogenic sarcoma that impinged on her airway. The treatment and postoperative course will be discussed in detail.

## 1. Introduction

Malignant odontogenic tumors are classified as odontogenic carcinomas and odontogenic sarcomas. The recent World Health Organization (WHO) classification grouped all the histological variants of sarcomas that arise from tooth-forming apparatus into one broad category: odontogenic sarcoma or amelobalstic fibrosarcoma [1]. Odontogenic sarcoma can be further classified as primary or secondary. Primary odontogenic sarcoma arises de novo without any preexisting lesion; however, secondary odontogenic sarcoma arises as a malignant transformation of the previous lesion, particularly the Ameloblastic Fibroma [2,3].

Odontogenic sarcomas are exceedingly rare and account for less than 5% of all Maxillofacial Sarcomas [1,4]. Their cancer biology is poorly understood, and their histopathological features overlap with fibrosarcoma and other spindle-cell sarcomas, making them very difficult to diagnose. Moreover, the epithelial components of odontogenic sarcomas will gradually disappear and become sparse as the malignant transformation progresses, which makes the diagnosis difficult. It usually affects the younger population. The posterior mandible is the most common site affected. Radiographically, it appears as a large destructive radiolucent lesion with ill-defined margins [3,5,6].

The diagnosis of an odontogenic sarcoma is made histopathologically. It is a true mixed tumor with benign epithelial components and sarcomatous changes of the ectomesenchymal cells [7]. The mesenchymal component of an odontogenic sarcoma shows cellular atypia, increased cellularity, mitosis, and a palisading pattern with a possible stepwise progression from a benign to malignant disease [8].

## 2. Case Report

The aim of this article is to describe the case of a 30-year-old African American female patient who presented at the Emergency Department (ED) in May 2023 at the University of Texas Medical Branch. As per the institution’s policy, IRB approval is not required for case reports, and photographic and informed consent was obtained from the patient to publish her case in a scientific journal, with progressive dyspnea, dysphagia, difficulty managed oral secretions, and decreased oral intake for the last three weeks. The family reported that she had increased swelling for the previous month, which was thought to be an odontogenic infection. The patient was seen by another ED-provider five weeks before and was discharged home with Amoxicillin-Clavulanic acid 875 mg PO every 12 h for 7 days. Oral and Maxillofacial Surgery services were consulted for urgent evaluation. Upon examination, the patient had a large expansile ulcerative mass intraorally, extending from the left posterior mandible and completely obliterating the oropharynx (Figure 1). The patient could not lay flat and was on 4 L of Oxygen via a nasal cannula. The patient denied any medical problems, taking any medications, or any allergy to any medicines. Based on the initial examination, the decision was to proceed with an awake tracheostomy to secure the airway, direct laryngoscopy, and incisional biopsy of the mass to establish the diagnosis. The procedure was performed uneventfully, and the patient had a CT Maxillofacial and Neck with an IV contrast. The scan showed a massive, expansile mass originating from the left posterior mandibular body, invading the pharynx’s posterior wall, with a complete obliteration of the oropharynx (Figure 2). Also, there was a highly suspicious enlarged lymph node at level 1B (Figure 3).

The biopsy of the oral cavity mass was sent to surgical pathology. Received in surgical pathology were multiple fragments of white-tan soft tissue. Histological evaluation revealed hypercellular spindle cells with surface ulceration and areas of necrosis. The spindle cells were of stellate morphology with myxoid background (Figure 4). No epithelial component was identified in the biopsy. The cellular morphology of the spindle cells and the myxoid background resembled ameloblastic fibroma. Still, they were more cellular with the presence of numerous mitoses (10 per 10 high-power fields) and areas of necrosis (Figure 5), together, with the destructive radiological features and the close association with mandibular teeth, the lack of the epithelial component, a diagnosis of high-grade odontogenic sarcoma was rendered.

The case was discussed at the head and neck tumor board, and the decision was to obtain a PET-CT scan and start the patient on chemotherapy before surgery. The PET-CT scan was negative for systemic metastasis. The patient started on the first cycle of chemotherapy with the AIM regimen, including Doxorubicin, Ifosfamide, and Mesna. The patient finished the first chemotherapy cycle in 7 days. However, she suffered from severe intractable nausea and vomiting and could not tolerate the chemotherapy. Also, the patient received three units of RBC due to the severe anemia that developed due to the chemotherapeutic regimen. Furthermore, based on radiologic evaluation, the mass did not respond to chemotherapy. The case was discussed again in the multidisciplinary board meeting, and the decision was to discontinue the chemotherapy and proceed with surgery once the patient recovered. 

During recovery from chemotherapy, surgical planning (VSP) was used to plan the surgery. Based on the computer simulation, the planned surgery was to achieve 3 cm margins of bone and soft tissue around the tumor. Also, VSP was used to design the reconstruction using an osteocutanous free fibula flap with dental implant placement (Figure 6A,B). After about five weeks, the patient underwent a hemi mandibulectomy with disarticulation, lateral pharyngectomy, selective neck dissection (level I-III), and immediate reconstruction using an osteocutaneous fibula flap (Figure 7A,B and Figure 8).

Received in surgical pathology was a composite hemimandibulectomy specimen with a polypoid exophytic mass closely associated with tooth 17–22. The mass measured 13 × 11 × 6 cm (Figure 9A–C). With a partial response to chemotherapy, more areas with necrosis were noticed. The cellularity was decreased compared to the original biopsy. A minute component ameloblastic epithelial component was identified with a positive immunostain of CK-19, a reliable marker of ameloblastic origin. Scattered hypercellular areas were seen with numerous mitoses and an aberrant expression of p53. All resection margins were negative, with 19 negative lymph nodes from levels IIA, IIB, and III neck dissections. The final pathological stage was ypT3 pN0. Additional molecular studies showed the neoplastic cells are negative for BRAF, NRAS, and KRAS mutations (Figure 10A–D).

The final result of the case was discussed again in the head and neck tumor board, and the consensus was to proceed with post-operative radiotherapy. The patient has completed the radiation therapy and is now six months with no evidence of the disease Figure 11.

The patient continued to follow-up every 3 months for the first year, and she is now 9 months on with no evidence of relapse. We will obtain a PET/CT at 1 year follow-up as per the institutional tumor-board recommendations.

## 3. Discussion

Odontogenic sarcomas are extremely rare, with only 100 cases reported in the English literature. The WHO simplified the classification system in 2017, and odontogenic sarcoma covers a group of mixed odontogenic tumors in which only the mesenchymal component shows malignant features [9]. In this article, we present a case of primary ameloblastic sarcoma with very rapid growth with an obliteration of the oropharynx, resulting in an emergency airway situation.

A recent systematic review showed that 24.7% of primary odontogenic sarcoma were misdiagnosed as benign odontogenic lesions [10]. Therefore, obtaining adequate biopsy and examination of the histopathological slides by an experienced pathologist is highly recommended.

Odontogenic sarcomas are rare, and careful histological examination is vital to establish the diagnosis. The epithelial component is usually benign with variables but not more than mild atypia. On the other hand, hypercellularity, nuclear atypia, high mitotic activity, and the overgrowth of the mesenchymal/stromal component are characteristic. Those features are indistinguishable from other spindle-cell sarcomas [11]. Fibrosarcoma and spindle-cell sarcoma are part of the differential diagnosis of the odontogenic sarcoma. The main striking difference is the lack of ameloblastic epithelium in the former and the presence of ameloblastic fibroma in the latter. Nonetheless, several reports in the literature documented that the epithelial components of odontogenic sarcoma will gradually disappear or become extremely sparse as the malignant transformation progresses, as occurred in this case. Therefore, we speculate that multiple reports of fibrosarcoma of the mandible could have been an odontogenic sarcoma with a disintegration of the odontogenic epithelium [12]. It is known that odontogenic sarcomas may arise from the malignant transformation of ameloblastic fibromas. Histopathologically, the ameloblastic fibroma shows no malignant features compared to odontogenic sarcomas. It is generally hypothesized that a malignant transformation from an ameloblastic fibroma to a secondary odontogenic sarcoma occurs stepwise. Also, this stepwise malignant transformation may be associated with chromosomal instability [2,3,13].

Immunohistochemical markers are used to establish the diagnosis of odontogenic sarcoma. Cytokeratin can identify epithelial nests and rule out sarcomas. Cells in the sarcomatous area are positive for CD34, vimentin, p53, and Ki 67; however, they are negative for S-100, smooth muscle actin, desmin, CD 68, and CD 117 [5,14,15], but these immunohistochemical stains are not very informative other than of values to rule out other malignancies.

Molecular analysis has shown a predominant BRAF-V600E mutation in odontogenic sarcomas.

Recent molecular analysis studied seven cases of odontogenic sarcomas and found five out of seven cases to be positive for the BRAF-V600E mutation. Targeted therapy for the BRAF-V600E mutation is currently available, and it would be an excellent option for patients with this aggressive, rare disease [12]. In this case, the BRAF V600E immunostain was performed and was negative.

Due to the rarity of the disease, there are no established guidelines regarding staging and treatment recommendations. Surgery is the mainstay treatment for resectable tumors. Neck dissection for the N0 neck is not indicated. Induction chemotherapy is controversial, but the consensus is that it is not beneficial unless there is a nodal involvement. Post-operative adjuvant therapy is indicated for positive/close margins. However, the benefit of post-operative radiotherapy for clear margins is debatable [3,16,17,18]. Chrcanovic et al. [3] found in their systematic review that patients who had chemotherapy and radiation therapy had higher recurrence rates than those who did not. However, it was not clear if the patients included received the chemotherapy and radiation therapy as adjuvant or neo-adjuvant. In this case, after an extensive discussion at the institutional multidisciplinary head and neck tumor board, the decision was to proceed with induction chemotherapy mainly because of the possible nodal involvement that was shown during the CT scans and due to the extensive size of the mass. The medical oncologist started the patient on ifosfamide and doxorubicin based on the previously reported good response from Gatz et al. [19]. However, there was no response in our patient after the completion of the chemotherapy cycle, and the patient suffered from severe thrombocytopenia and leukopenia, which delayed the planned surgery for three more weeks until the patient recovered.

Now, with the advances in virtual surgical planning (VSP), reconstructive surgeons are able to perform complex reconstructions with great precision and excellent outcomes. In our case, VSP was used to design the reconstruction of the hemi mandibulectomy defect with a condylar reconstruction using one of the fibula segments. Initially, we planned to perform two microvascular flaps: a fibula flap to reconstruct the mandible and a radial forearm flap to reconstruct the lateral pharynx. However, we were able to harvest the soleus muscle perforator flap as a chimeric flap from the peroneal artery. The muscle adapted well to the lateral pharyngeal wall and mucosalized fully within two weeks. Also, with the advances in jaw reconstruction, we could place two dental implants to help with dental rehabilitation. Finally, the sensory-nerve reconstruction of the inferior alveolar and lingual nerves using a cadaveric nerve graft is feasible in benign tumors. In this case, we opted not to perform microneural reconstruction for both nerves in anticipation of postoperative radiation therapy and the lack of evidence to support the use of cadaveric nerves in those circumstances.

The prognosis of odontogenic sarcomas is not known. The recurrence rate is high, at up to 24.5%. Also, about 5.5% of patients suffered a distant metastatic deposit in the lungs. There are several reports in the literature of patients who suffered from multiple recurrences and died from the disease [8,13,20]. A survival study conducted in 2016 studied all subjects with malignant odontogenic tumors (carcinomas and sarcomas) and found that patients who did not receive radiation therapy were 20 times more likely to die from cancer [20]. Therefore, post-operative radiation therapy is generally recommended for odontogenic sarcoma. In summary, survival studies have shown that an increase in the size of the tumor, old age, and a lack of post-operative radiation are statistically significant factors with a negative impact on survival [9,20].

## 4. Conclusions

An odontogenic sarcoma is a rare head and neck sarcoma that can arise de novo or from the benign ameloblastic fibroma counterpart. Accurate histopathological examination by a head and neck pathologist is essential to establish the diagnosis. Surgery is the primary treatment. Induction chemotherapy has shown no benefit. Surgical resection with a negative margin and post-operative radiotherapy are the keys to improved survival. Long-term follow-up is recommended.

## Figures and Tables

**Figure 1 biomedicines-12-00615-f001:**
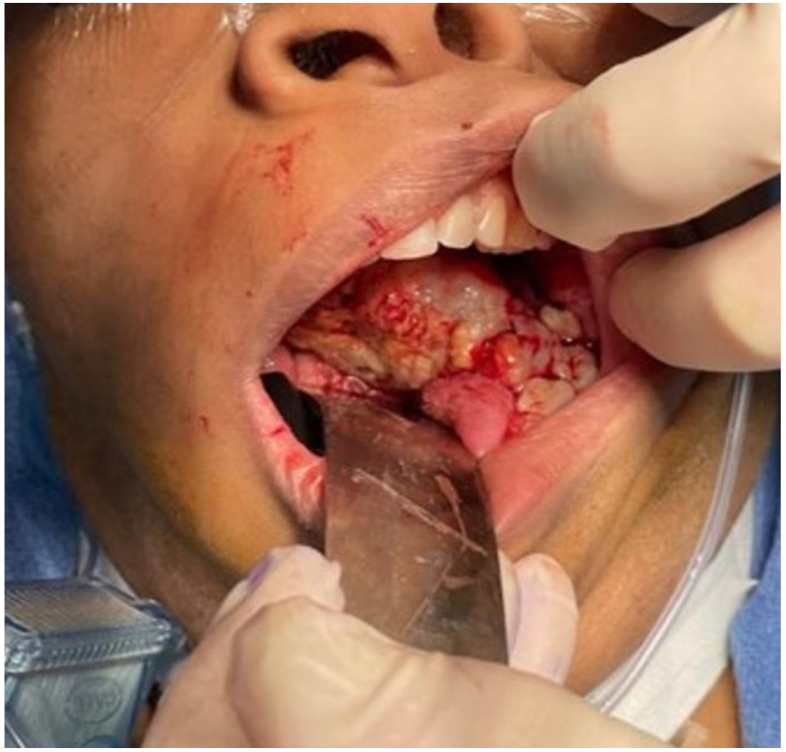
Intraoral mass extending from the left posterior mandible and completely obliterating the airway.

**Figure 2 biomedicines-12-00615-f002:**
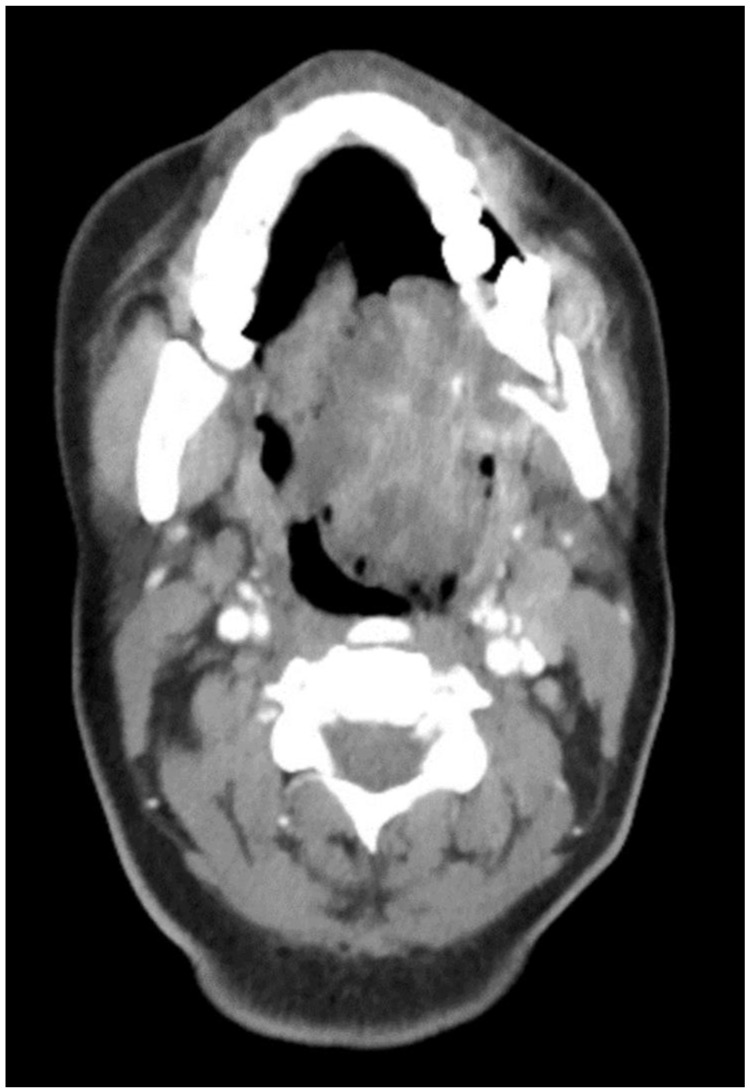
CT Scan with contrast showing the mass extending to the oropharynx with complete airway obstruction.

**Figure 3 biomedicines-12-00615-f003:**
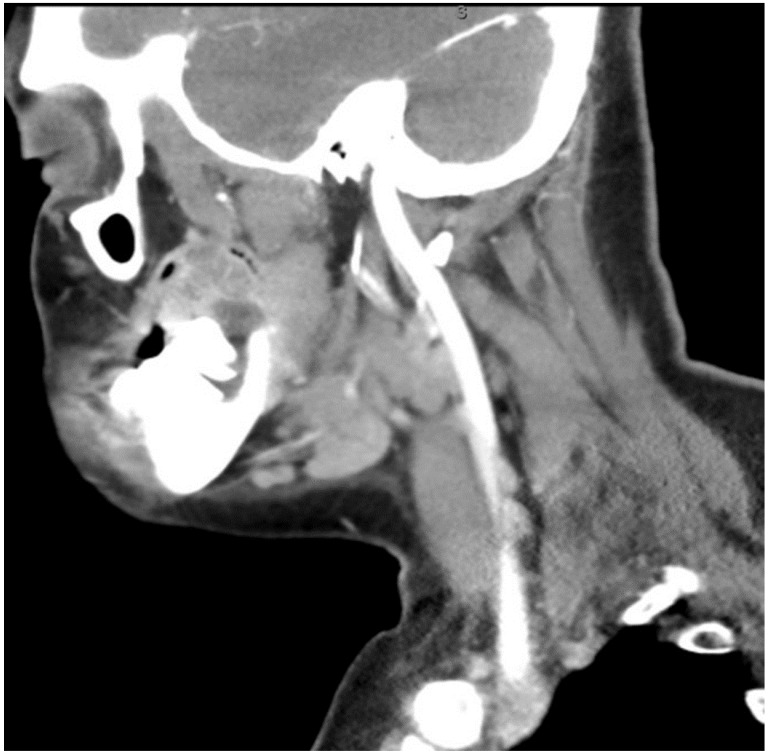
CT neck with contrast showing the enlarged lymph node at level 1B.

**Figure 4 biomedicines-12-00615-f004:**
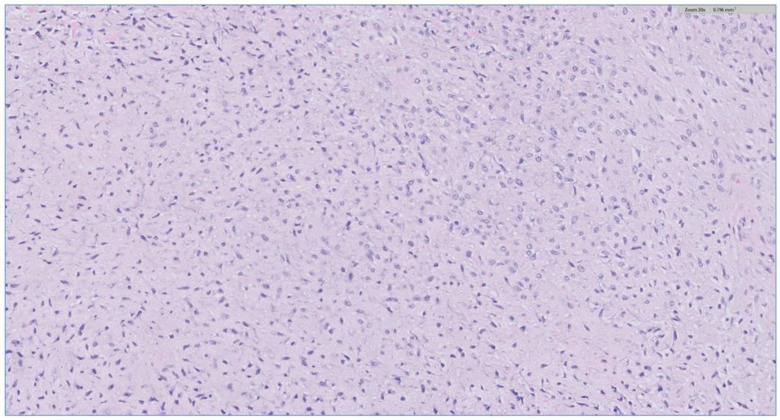
The H&E stain of the incisional biopsy shows that the spindle cells are of stellate morphology with a myxoid background.

**Figure 5 biomedicines-12-00615-f005:**
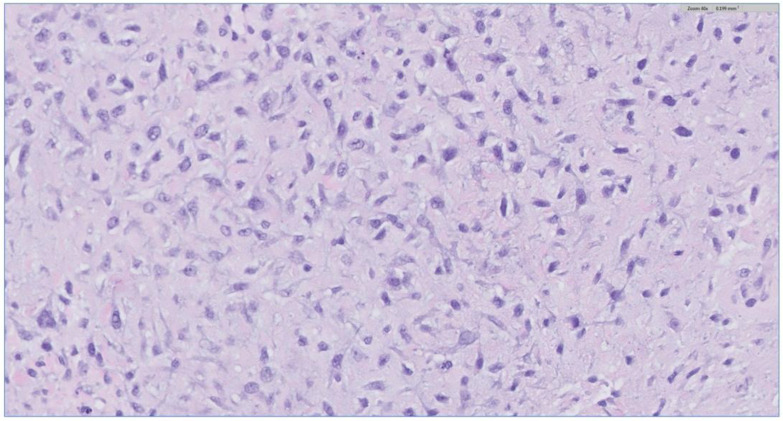
H&E stain of the incisional biopsy shows the cellular component of the mass, extensive mitosis, and areas of ulceration.

**Figure 6 biomedicines-12-00615-f006:**
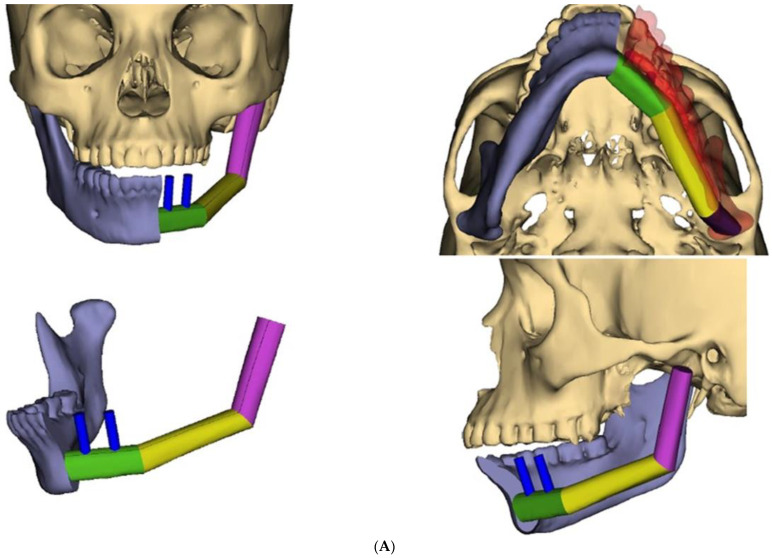
(**A**) Virtual surgical planning for the resection and reconstruction. The mass was exerting pressure on the occlusion, and the patient occlusion had to be readjusted. (**B**) Illustration showing the extent of the mass to the tongue, mandible, and oropharynx. Noted the dashed line for the planned hemi-mandibulectomy according to the VSP.

**Figure 7 biomedicines-12-00615-f007:**
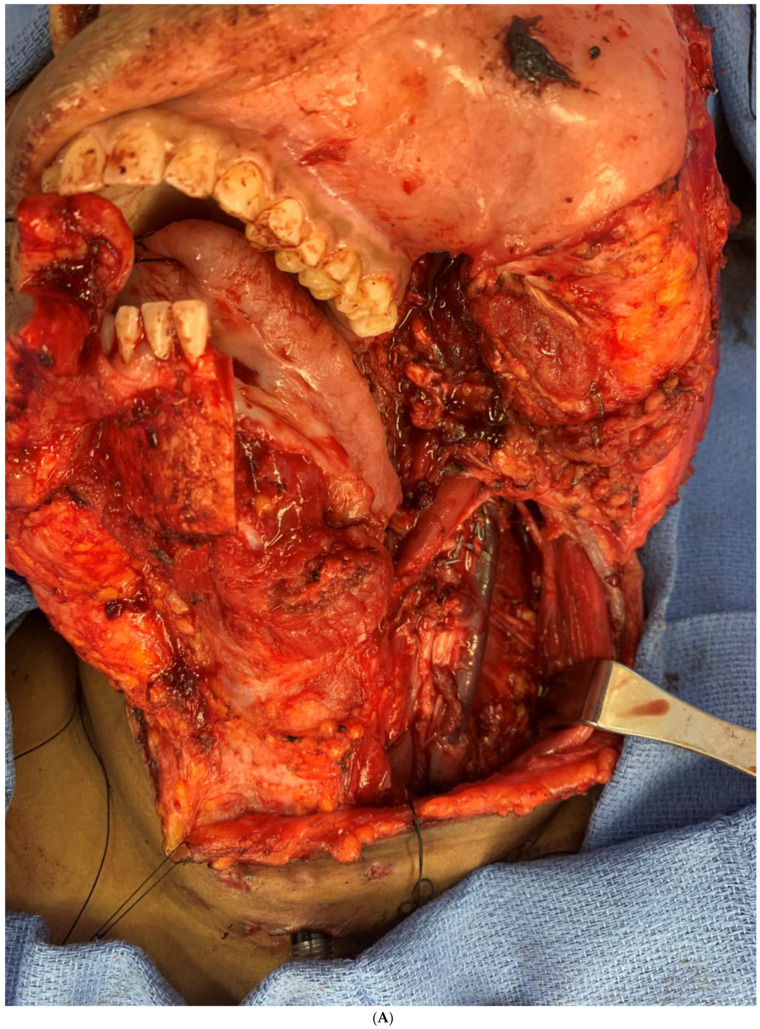
(**A**) Intraoperative images showing the surgical defect after the hemimandibulectomy and neck dissection. (**B**) The illustration shows the defect’s extent after the hemimandibulectomy, lateral pharyngectomy, and neck dissection.

**Figure 8 biomedicines-12-00615-f008:**
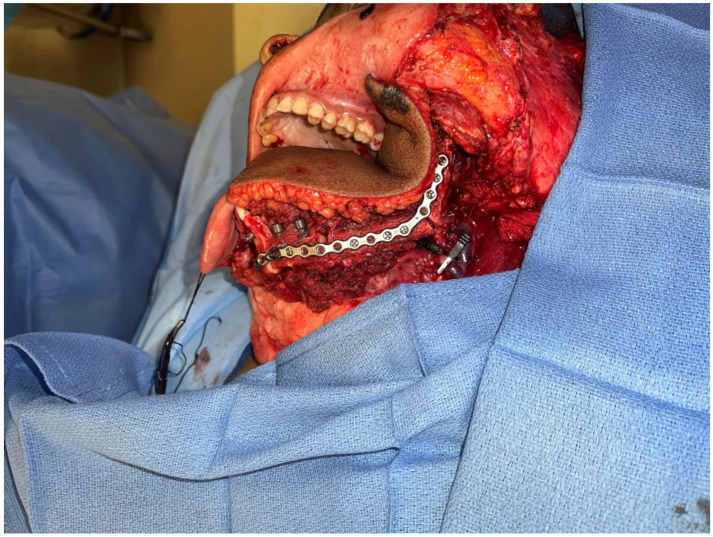
Intraoperative images showing the inset of the osteocutaneous fibula flap with dental implant placement to help in future dental rehabilitation.

**Figure 9 biomedicines-12-00615-f009:**
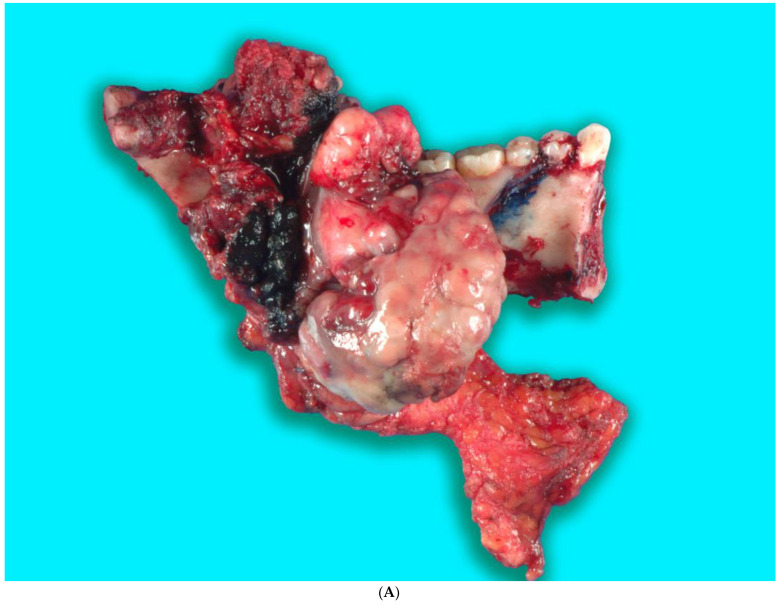
(**A**,**B**). Gross specimen for hemimandibulectomy, lateral pharyngectomy, and neck dissection. (**C**) The mass is bisected and shows the necrotic area of the tumor at the periphery.

**Figure 10 biomedicines-12-00615-f010:**
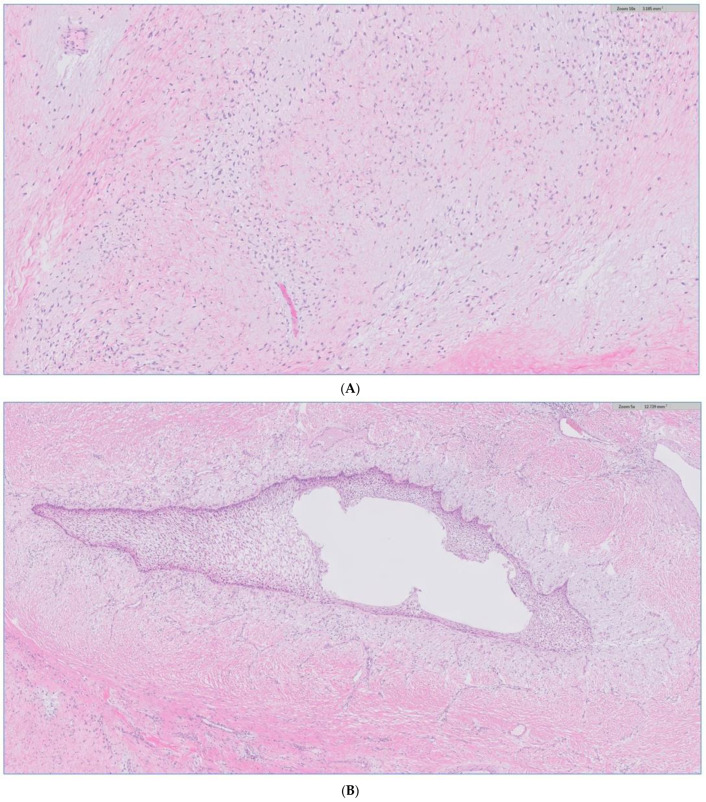
(**A**) H&E stain of the incisional biopsy shows the cellular component of the mass, extensive mitosis, and areas of ulceration. (**B**) A minute component of ameloblastic epithelial is identified within the tumor in the final pathology. (**C**) positive immunostain of CK-19, a reliable marker of ameloblastic origin. (**D**) Scattered hypercellular areas are seen with numerous mitoses and aberrant expression of p53.

**Figure 11 biomedicines-12-00615-f011:**
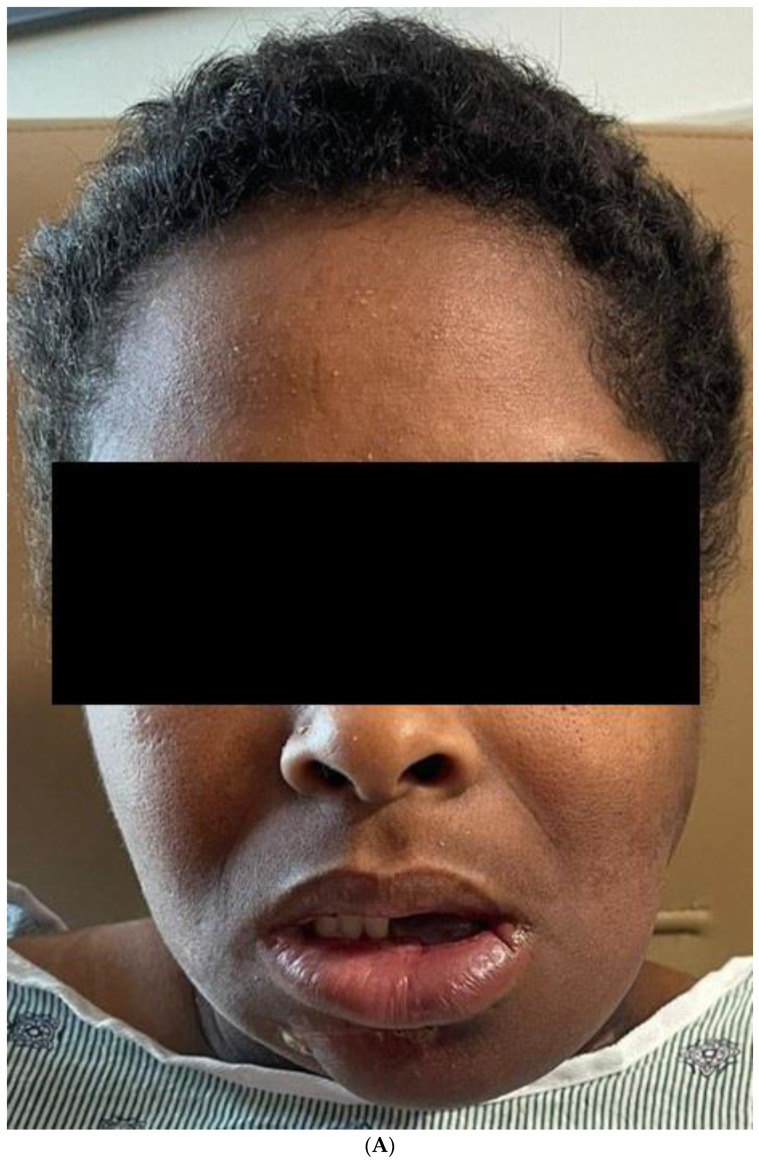
(**A**) Frontal and profile picture of the patient 6 months after finishing the therapy. (**B**) Intraoral picture of the flap.

## Data Availability

The study only included one single case. Data was not collected.

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
