# Peer review of "Odontogenic Sarcomas of the Mandible"

_biomedicines, 2024, doi:10.3390/biomedicines12030615_

Round 1

Reviewer 1 Report

Comments and Suggestions for Authors

This is a case report on Odontogenic Sarcoma of the Mandible.

In addition, the available literature and evidence is briefly summarized.

The case is clearly presented, with well-structured references to the available literature.

This report is of interest to those caring for patients with odontogenic tumors, as odontogenic sarcoma is an exceedingly rare condition and there is few if any guidance in the existing literature.

Thus, publication of this case seems worth while.

However, there are two minor suggestions to the authors:

I would appreciate if the abstract would at least very briefly refer to the actual case presented, in addition to more general remarks on odontogenic sarcoma, e.g. “Here we present….”

Moreover, while the case reported certainly is very interesting and challenging, follow-up as reported is very short. The multidisciplinary treatment planning and discussion are important and relevant, and the treatment path chosen seems appropriate. However, a follow-up of only six months after treatment seems rather short in sarcoma, which may relapse late. Maybe by now more time has passed and the authors can give an update on follow-up?

Author Response

Response to Reviewer:

1- This is a case report on Odontogenic Sarcoma of the Mandible.In addition, the available literature and evidence is briefly summarized.The case is clearly presented, with well-structured references to the available literature.This report is of interest to those caring for patients with odontogenic tumors, as odontogenic sarcoma is an exceedingly rare condition and there is few if any guidance in the existing literature.Thus, the publication of this case seems worth while.However, there are two minor suggestions to the authors:

I would appreciate if the abstract would at least very briefly refer to the actual case presented, in addition to more general remarks on odontogenic sarcoma, e.g. “Here we present….”

Response: We appreciate the reviewer taking the time and effort to review our manuscript and work on its improvement. The abstract is edited as suggested.

2- Moreover, while the case reported certainly is very interesting and challenging, follow-up as reported is very short. The multidisciplinary treatment planning and discussion are important and relevant, and the treatment path chosen seems appropriate. However, a follow-up of only six months after treatment seems rather short in sarcoma, which may relapse late. Maybe by now more time has passed and the authors can give an update on follow-up? 

Response: We absolutely agree with the reviewer. A 9-month follow-up was included with a future follow-up plan.

Reviewer 2 Report

Comments and Suggestions for Authors

The authors have presented this report of a case on an odontogenic sarcoma lesion in the left posterior mandible site in a 30 year old black female patient. Overall, I believe that the authors have presented an interesting case for the literature on a lesion that is very rare and has many unique features with a challenging diagnosis. However, there are some major issues with this manuscript  most importantly the absence of all of the figures both in the manuscript and in their online submission. Moreover, the authors need to conduct an extensive electronic search in the literature since a lot of their key data in their introduction either have no support or have extremely old references. I look forward to reviewing this paper once again after the authors have provided their figures in the manuscript and have addressed all of concerns and questions listed below:

Affiliations:

First of all, the authors list and affiliations do not comply with the required structure of the MDPI journals. The authors must provide a list of affiliations and must refer to each one with a number next to the name of each author. 

Abstract:

The authors have only given a short background on the occurrence and most affected sites of odontogenic sarcoma. While I appreciate the authors’ efforts to give a short introduction on their investigated pathology, I believe it would be way more appropriate to also give a short history and case presentation of the investigated case of this paper.

Introduction:

Lines 39, 40, and 41:

“The recent World Health Organization (WHO) classification grouped all the histological variants of sarcomas that arise from tooth-forming apparatus into one broad category: odontogenic sarcoma or amelobalstic fibrosarcoma1.” 

The authors have referred to reference number 1 as their source for this statement, however, they have failed to mention the year of publication for reference number 1. I looked up that article in the literature and the publication date is 2020. This article is a report of a case combined with a systematic review on the subject. But I do not understand the rationale behind choosing this study as the main and only reference for this statement. The authors mention the “recent WHO classification for odontogenic sarcoma” yet they refer to an independent study on a case report from 2020? While I have found a published classification for Odontogenic Sarcoma from WHO published in 2022 titled: “The World Health Organization Classification of Odontogenic Lesions: A Summary of the Changes of the 2022 (5th) Edition”. In this report all the new changes made to their WHO classification have been listed which I absolutely encourage the authors to fully examine this report and use it as part of their support and reference for their statements. 

Lines 41, 42, 43, 44, and 45:

“Odontogenic sarcoma can be further classified as primary and secondary. Primary odontogenic sarcoma arises de novo without any preexisting lesion; however, secondary odontogenic sarcoma arises as a malignant transformation of the previous lesion, particularly the Ameloblastic Fibroma2.”

The authors have used their reference (number 2) which was published in 2012 as their support for this claim. As the authors have mentioned themselves, oral and maxillofacial pathological lesions’ classifications are always being reevaluated and constantly changing due to their complicated and inter-lesion-connected nature. Therefore, authors and clinicians that tend to work on research projects related to pathological lesions, must constantly update their knowledge and only use the newest publish articles for basic data on their investigated lesions. Hence, talking about the primary and secondary classifications of odontogenic sarcoma while referring to a 2012 published study is unacceptable. I highly suggest authors update their extensive electronic search on “odontogenic sarcoma” and only use the newest published materials on their subject regarding its diagnosis, prognosis, classifications and treatment protocols.

Line 47:

“Their cancer biology is poorly understood” I kindly ask the authors to elaborate on this statement, what exactly do you mean by poorly understood? If it is in relation with other pathologies then the authors must name all of the other similar lesions. 

Line 48:

“making it very difficult to diagnose” as I mentioned above, the authors need to elaborate on their statements regarding the challenging nature and unique features of odontogenic sarcoma. And here that the authors claim that the diagnosis of odontogenic sarcoma is very challenging and difficult, they must specify why? The authors must name which other pathological lesions are being considered for differential diagnosis next to odontogenic sarcoma. 

I believe that the second paragraph of the introduction has mentioned a lot of key data regarding the epidemiologic status of Odontogenic sarcoma by claiming that “It usually affects the younger population” and that “The posterior mandible is the most common site affected.” The authors have supported these very important claims by referring to reference number 4 which is a study from 2001. I believe that this is absolutely unacceptable. How are the readers supposed to trust your claims and statement when your only source of information extracted from the literature is from 2001? Even if none of these epidemiologic data has changed in the last two decades, the authors are still responsible to find the newest published studies for each and every one of their statements. Once again, I highly advise the authors to conduct a proper extensive electronic search in the literature with proper search terms in order to not miss out on any related article. 

Case Report: 

“A 3o-year-old African American female patient presented to the Emergency Department (ED) at the University of Texas Medical Branch with progressive dyspnea, dysphagia, difficulty managing oral secretions, and decreased oral intake for the last three weeks.” Kindly add the exact date in which the patient was admitted to the hospital. 

Nowhere in the manuscript was any evidence of the overall health status of the patient. Kindly include these data.

The authors have referred to multiple figures in their case report yet only their figure legends are available in the text. The figures are not available neither in the text nor in their submission. I am not able to fully evaluate this submission without having access to the figures of their experiment. Kindly provide these figures in your revised manuscript.

Line 62:

“was discharged home with an oral antibiotic” kindly specify the exact type of antibiotic that the patient was prescribed along with its dosage and total consumption days.

Author Response

  • The authors have presented this report of a case on an odontogenic sarcoma lesion in the left posterior mandible site in a 30 year old black female patient. Overall, I believe that the authors have presented an interesting case for the literature on a lesion that is very rare and has many unique features with a challenging diagnosis. However, there are some major issues with this manuscript . Moreover, the authors need to conduct an extensive electronic search in the literature since a lot of their key data in their introduction either need more support or have extremely old references. I look forward to reviewing this paper once again after the authors have provided their figures in the manuscript and have addressed all of the concerns and questions listed below:

Response: We appreciate the reviewer to take the time to review the manuscript. All figures were uploaded and attached to the original document. On the other hand, it was not the intention of the authors of this manuscript to provide a systematic review of the topic on hand, mainly due to the scarce information and the confusing terminology that is consistently changing. Nonetheless, we appreciate the reviewer for his suggestion, and we conducted another electronic research and updated our reference list to include more recent and relevant references as suggested. 

  • In my initial review of this paper, the figures were absent from the Word document. However, the editor has kindly provided a new file and I was able to review the figures too.
  • In pages 8 and 10 of the manuscript, the authors have used two figures showcasing a schematic illustration of the patient’s tumor and treatment plan before and after the surgery. Are these images original or have they been borrowed from other sources? If they are taken from other sources the authors must definitely refer to their reference and also obtain permission to use these materials in their paper.

Response: Thank you so much for taking the time to revise and review the manuscript with the images uploaded. All figures are original to the manuscript, and we have their copyright. 

  • In figure 11, the authors have shown a completely uncensored profile side picture of their patient. The authors must definitely mention in their text that they have received written consent forms signed by their patient to allow the authors to use her face without ant censoring in her eyes. Also, the authors must definitely include the ethical committee approval for this study in their text.

            Response: we have uploaded the written photographic consent signed by the patient. As per our institution's policy, case reports do not require ethical committee approval. We stated that in the case report description. 

Affiliations:

First of all, the authors list and affiliations do not comply with the required structure of the MDPI journals. The authors must provide a list of affiliations and must refer to each one with a number next to the name of each author. 

Response: This was updated and changed following the guidelines. 

Abstract:

The authors have only given a short background on the occurrence and most affected sites of odontogenic sarcoma. While I appreciate the authors’ efforts to give a short introduction on their investigated pathology, I believe it would be way more appropriate to also give a short history and case presentation of the investigated case of this paper.

Response: Thank you for the valuable suggestion. The abstract is edited as suggested.

Introduction:

Lines 39, 40, and 41:

“The recent World Health Organization (WHO) classification grouped all the histological variants of sarcomas that arise from tooth-forming apparatus into one broad category: odontogenic sarcoma or amelobalstic fibrosarcoma1.”

The authors have referred to reference number 1 as their source for this statement; however, they have failed to mention the year of publication for reference number 1. I looked up that article in the literature, and the publication date is 2020. This article is a report of a case combined with a systematic review on the subject. However, I do not understand the rationale behind choosing this study as the main and only reference for this statement.

Response: Thank you for your valuable suggestion. The reference was updated. The reason why we chose this reference is because it provides a comprehensive description of the disease following the new WHO nomenclature for odontogenic sarcoma. When we looked into the literature about Odontogenic Sarcoma, most of the literature used the old terminology of Odontogenic Fibrosarcoma. Therefore, that particular reference is relatively new and pertinent to our case. Moreover, this reference provides an extensive histopathological examination that helped our H&N Pathologist in establishing and confirming the diagnosis. 

  •  While I have found a published classification for Odontogenic Sarcoma from WHO published in 2022 titled: “The World Health Organization Classification of Odontogenic Lesions: A Summary of the Changes of the 2022 (5th) Edition”. In this report all the new changes made to their WHO classification have been listed which I absolutely encourage the authors to fully examine this report and use it as part of their support and reference for their statements.

Response: Thank you so much for suggesting that article. When we looked into the literature, the article published in the Turkish Journal of Pathology “The World Health Organization Classification of Odontogenic Lesions: A Summary of the Changes of 2022 (5th) Edition” was one of the first that showed up in our search. However, when we read the article; it didn’t provide any extensive description of the disease. Actually, there was only 1 paragraph the authors wrote to describe the odontogenic sarcoma. Please see the screenshot below. 

  • Lines 41, 42, 43, 44, and 45:

“Odontogenic sarcoma can be further classified as primary and secondary. Primary odontogenic sarcoma arises de novo without any preexisting lesion; however, secondary odontogenic sarcoma arises as a malignant transformation of the previous lesion, particularly the Ameloblastic Fibroma2.”

The authors have used their reference (number 2) which was published in 2012 as their support for this claim. As the authors have mentioned themselves, oral and maxillofacial pathological lesions’ classifications are always being reevaluated and constantly changing due to their complicated and inter-lesion-connected nature. Therefore, authors and clinicians who tend to work on research projects related to pathological lesions, must constantly update their knowledge and only use the newest published articles for basic data on their investigated lesions. Hence, talking about the primary and secondary classifications of odontogenic sarcoma while referring to a 2012 published study is unacceptable. I highly suggest authors update their extensive electronic search on “odontogenic sarcoma” and only use the newest published materials on their subject regarding its diagnosis, prognosis, classifications, and treatment protocols.

Response: Thank you so much for your valuable input. We have updated the reference list and included a couple more new references. 

  • Line 47:

“Their cancer biology is poorly understood” I kindly ask the authors to elaborate on this statement, what exactly do you mean by poorly understood? If it is in relation with other pathologies then the authors must name all of the other similar lesions.

  • Line 48:

“making it very difficult to diagnose” as I mentioned above, the authors need to elaborate on their statements regarding the challenging nature and unique features of odontogenic sarcoma. And here that the authors claim that the diagnosis of odontogenic sarcoma is very challenging and difficult, they must specify why? The authors must name which other pathological lesions are being considered for differential diagnosis next to odontogenic sarcoma.

Response: Thank you for the input. Both statements were explained in detail in the third paragraph of the discussion section. The reason we included this in the introduction is to “hook” the reader and engage them with the manuscript to read it. Nonetheless,  A statement describing the difficulty and challenges in diagnosis is added to the manuscript. 

  • I believe that the second paragraph of the introduction has mentioned a lot of key data regarding the epidemiologic status of Odontogenic sarcoma by claiming that “It usually affects the younger population” and that “The posterior mandible is the most common site affected.” The authors have supported these very important claims by referring to reference number 4, which is a study from 2001. I believe that this is absolutely unacceptable. How are the readers supposed to trust your claims and statement when your only source of information extracted from the literature is from 2001? Even if none of these epidemiologic data has changed in the last two decades, the authors are still responsible to find the newest published studies for each and every one of their statements. Once again, I highly advise the authors to conduct a proper extensive electronic search in the literature with proper search terms in order to not miss out on any related article.
  • Response: Thank you for the valuable review. It was never the intent of the authors to conduct a systematic review of the topic. Nonetheless, an updated list of references was added and highlighted in the reference section. 

  • “A 3o-year-old African American female patient presented to the Emergency Department (ED) at the University of Texas Medical Branch with progressive dyspnea, dysphagia, difficulty managing oral secretions, and decreased oral intake for the last three weeks.” Kindly add the exact date in which the patient was admitted to the hospital.

Response: This was added

  • Nowhere in the manuscript was any evidence of the overall health status of the patient. Kindly include these data.

Response: Thank you for pointing this out. This was added 

  • The authors have referred to multiple figures in their case report yet only their figure legends are available in the text. The figures are not available neither in the text nor in their submission. I am not able to fully evaluate this submission without having access to the figures of their experiment. Kindly provide these figures in your revised manuscript.

Response: This was fixed by the publisher. Thank you for the help Ms. Butica 

  • Line 62:

“was discharged home with an oral antibiotic” kindly specify the exact type of antibiotic that the patient was prescribed along with its dosage and total consumption days.

Response: This was added